# 3D-Printed Piezoelectric Porous Bioactive Scaffolds and Clinical Ultrasonic Stimulation Can Help in Enhanced Bone Regeneration

**DOI:** 10.3390/bioengineering9110679

**Published:** 2022-11-11

**Authors:** Prabaha Sikder, Phaniteja Nagaraju, Harsha P. S. Naganaboyina

**Affiliations:** Department of Mechanical Engineering, Cleveland State University, Cleveland, OH 44115, USA

**Keywords:** piezoelectric, electroactive, bioactive, orthopedics, 3D printing, ultrasonic stimulation

## Abstract

This paper presents a comprehensive effort to develop and analyze first-of-its-kind design-specific and bioactive piezoelectric scaffolds for treating orthopedic defects. The study has three major highlights. First, this is one of the first studies that utilize extrusion-based 3D printing to develop design-specific macroporous piezoelectric scaffolds for treating bone defects. The scaffolds with controlled pore size and architecture were synthesized based on unique composite formulations containing polycaprolactone (PCL) and micron-sized barium titanate (BaTiO_3_) particles. Second, the bioactive PCL-BaTiO_3_ piezoelectric composite formulations were explicitly developed in the form of uniform diameter filaments, which served as feedstock material for the fused filament fabrication (FFF)-based 3D printing. A combined method comprising solvent casting and extrusion (melt-blending) was designed and deemed suitable to develop the high-quality PCL-BaTiO_3_ bioactive composite filaments for 3D printing. Third, clinical ultrasonic stimulation (US) was used to stimulate the piezoelectric effect, i.e., create stress on the PCL-BaTiO_3_ scaffolds to generate electrical fields. Subsequently, we analyzed the impact of scaffold-generated piezoelectric stimulation on MC3T3 pre-osteoblast behavior. Our results confirmed that FFF could form high-resolution, macroporous piezoelectric scaffolds, and the poled PCL-BaTiO_3_ composites resulted in the d_33_ coefficient in the range of 1.2–2.6 pC/N, which is proven suitable for osteogenesis. In vitro results revealed that the scaffolds with a mean pore size of 320 µm resulted in the highest pre-osteoblast growth kinetics. While 1 Hz US resulted in enhanced pre-osteoblast adhesion, proliferation, and spreading, 3 Hz US benefited osteoblast differentiation by upregulating important osteogenic markers. This study proves that 3D-printed bioactive piezoelectric scaffolds coupled with US are promising to expedite bone regeneration in orthopedic defects.

## 1. Introduction

Piezoelectric materials are defined to be “smart” or “electroactive” materials that generate electrical fields (EFs) or electrical stimuli in response to mechanical deformations. The deformations cause an asymmetric shift of ions or charges in the materials, which induces a change in the electrical polarization, thus generating EFs [1]. Piezoelectric materials are widely used in electronic applications such as transducers, sensors, and actuators, but in the past decade, there has been a growing interest in using them as an implant or scaffold for tissue regeneration [2]. One of the primary reasons to drive this interest is the capability of piezoelectric materials to provide therapeutic EFs to cells and tissues without the need for additional power sources and wire connections. The EFs are beneficial for tissue regeneration, as they influence essential cell functions such as cell proliferation and migration [3]. In orthopedics, the applied EFs simulate bioelectrical cues of natural living bone and regulate metabolic activities such as bone growth, structural remodeling, and fracture healing [4]. Moreover, physiological compressive loads have been proven to increase the negative charges in a fractured bone region due to the bone’s innate piezoelectric potential, which helps promote matrix mineralization, osteoblast functions, and bone regeneration [5,6]. Hence, sufficient evidence indicates that applied piezoelectric stimulation can help improve bone regeneration.

Previous studies have confirmed that the stimulation or the EF generated from electroactive piezoelectric implants can help expedite bone fracture repair [7,8]. Such electroactive scaffolds develop EFs with the application of physiological loads at the implantation site, which is usually the region of the bone defect. Precisely, the hyperpolarization from the piezoelectric effect activates voltage-gated calcium channels of the cell membrane, which helps in osteoblast proliferation, resulting in new bone formation [7,9]. The piezoelectric scaffolds also develop surface electrical charges under external stress, similar to bone, and the polarized piezoelectric surfaces improve the osteogenic performance of natural bone. For instance, Liu et al. [10] developed electropositive ferroelectric bismuth ferrite (BiFeO_3_) nanofilms and observed that the films enhanced protein adsorption and mesenchymal stem cell attachment, spreading, and osteogenic differentiation. In the rat femur, the polarized BiFeO_3_ nanofilms in the presence of built-in electric fields resulted in rapid and superior osseointegration [10]. Similarly, Ribeiro et al. [11] implanted poled and un-poled *β-*poly(vinylidene fluoride) (*β-*PVDF) films in bone defects created in rats and observed significantly more defect closure and bone remodeling with poled *β-*PVDF films as opposed to the non-poled ones. Hence, it is well comprehended that the controlled application of electrical stimuli or EFs from implanted piezoelectric scaffolds can be an effective therapeutic tool in bone regeneration and consequentially treat bone defects expeditedly.

Generally, the piezoelectric scaffolds are developed from biomaterials such as piezo-bioceramics and piezo-biopolymers. Piezoelectric bioceramics have also been designed as in vivo energy harvesters for biosensors. The most well-known piezoelectric bioceramics are barium titanate (BaTiO_3_), boron nitride (BN), and magnesium silicate (MgSiO_3_) [12]. The most well-known piezo-biopolymers are PVDF and their copolymers, such as P(VDF-TrFE), and natural polymers, such as collagen and chitosan [12]. Moreover, various composites such as hydroxyapatite–BaTiO_3_ (HA-BaTiO_3_), PVDF–BaTiO_3_, and HA–MgSiO_3_–Chitosan have been developed to take advantage of the combined properties of the piezo-ceramics and polymers [8,12]. Yet, the majority of the piezoelectric scaffolds were manufactured using conventional techniques such as electrospinning, solvent casting, slip casting, solvent forming-particulate leaching, or freeze casting [12], which do not take into consideration the design-specific needs of the medical implant industry. Especially due to the high clinical demand for developing patient-specific therapies to augment treatment outcomes, of late, there is an imperative need to develop defect- and anatomy-specific regenerative implants.

Three-dimensional printing, or additive manufacturing, is a high-powered fabrication technique that can produce parts of any complex architecture, shape, and dimension layer by layer. In recent years, even though there has been a surge in the application of 3D printing to manufacture biomedical devices, very few reports have focused on developing bioactive and regenerative piezoelectric implants utilizing this process. The majority of the focus was on using 3D printing to develop biosensors, not bioactive and regenerative implants. [13]. For instance, Grinberg et al. [14] developed Polyamide-11 (PA-11) and BaTiO_3_ 3D-printable composite filaments and used the fused deposition modeling technique to print proof-of-concept models of piezoelectric sensors to monitor the applied force on the knee prosthesis. Furthermore, even though researchers used 3D printing to develop regenerative scaffolds, they utilized complex techniques. For instance, Jiang et al. [15] developed porous BaTiO_3_ scaffolds with the help of digital light processing (DLP) technology, but it required sintering as an additional processing step to develop the scaffolds. On the other hand, Polley et al. [16] used the binder jet 3D printing technique to develop porous HA-BaTiO_3_ scaffolds. Similarly, Yang et al. [17] used selective laser sintering (SLS) to create graphene and BaTiO_3_-based scaffolds. However, both SLS and binder jet involve long wait times after the printing process, processing complexities, and rigorous thermal-post treatment such as sintering.

On the contrary, “fused filament fabrication” (FFF) is a one-step, rapid and easy fabrication method to develop implants with favorable material properties. FFF can directly form robust scaffolds, which might not need sintering as an additional step [18,19,20,21,22]. Importantly, it involves much less material wastage than SLS or binder-jet printing. Our research group used the FFF to develop various design-specific, regenerative implants comprising novel and bespoke polymer–ceramic compositions [23,24].

Not much effort has been made to develop bioactive and biodegradable piezoelectric compositions. For instance, extensive efforts have been made to develop PVDF-based scaffolds [11,25,26,27], but PVDF is neither a bioactive nor biodegradable piezoelectric biomaterial. Poly (lactic acid-co-glycolic acid) (PLGA)–BaTiO_3_ composites [28] and PLLA–BaTiO_3_–graphene [17] have been developed, but PLGA and PLLA are known to degrade into acidic byproducts such as glycolic acid and lactic acid that might be harmful to healthy cells [29]. Graphene is also known to exhibit cytotoxicity in various kinds of cells [30].

Hence, to address some of the literature gaps, in this study, we developed a unique electroactive and bioactive polymer–ceramic composition in the form of 3D-printable filaments such that the filaments could be utilized in a FFF 3D printing setup to manufacture design-specific piezoelectric orthopedic scaffolds. To achieve that, first, we introduced micron-sized piezoelectric BaTiO_3_ particles in the polycaprolactone (PCL) polymer matrix by solvent-casting and subsequently used a customized extrusion process to form uniform-diameter 3D-printable PCL–BaTiO_3_ composite filaments. Then, the filaments served as feedstock material for an FFF setup to develop the design-specific, porous scaffolds. We chose PCL because it is an FDA-approved, biodegradable polymer that has been extensively used in implantable biomaterials, and we chose BaTiO_3_ because of its well-known piezoelectric and biocompatible properties. After the scaffold development, we performed thorough material, mechanical, and in vitro property analyses of the 3D-printed structures. The phase composition and morphology of the implants were determined by X-ray diffraction and scanning electron microscopy (SEM). Mechanical properties of the composites were determined using ASTM standards, and the proliferative and differentiative capacity of the scaffolds was assessed in vitro using MC3T3 pre-osteoblast cells.

We believe that the result of this proof-of-concept study is the first step toward developing an effective orthopedic-based regenerative medicine for the society. EFs have been used in clinics for wound healing for several years. Furthermore, the components/materials of the orthopedic scaffold used in this study are FDA-approved, hopefully making the regulatory route to be 510(k). More effective collaboration between clinician scientists, engineers, orthopedic surgeons, and federal funding agencies is required to conduct clinical trials and translate innovative medical devices such as this piezoelectric polymer–ceramic scaffold for expedited bone regeneration [31,32]. Notably, such scaffold-mediated expedited bone regeneration can be instrumental in the military community, where it is expected for the wounded soldiers to recover quickly and rejoin active duty. It can also benefit patients with diabetes with compromised or delayed wound healing and tissue regeneration rates [33].

## 2. Materials and Methods

### 2.1. Raw Materials

Polycaprolactone (PCL, molecular weight = 80,000) pellets, BaTiO_3_ (average particle size < 2 µm, molecular weight = 233.19) powder, and dichloromethane (CH_2_Cl_2_, anhydrous, ≥99.8%) were procured from Sigma Aldrich and used as is.

### 2.2. Composite Development

The entire process of developing the piezoelectric composites and the 3D-printable piezoelectric filaments is shown in Figure 1. First, PCL pellets were dissolved in dichloromethane following 15 % (*w*/*v*) concentration. Once the polymer was completely dissolved, a varying amount of BaTiO_3_ powder (25, 45, and 65 vol.%) was added to it, followed by a 3-step mixing procedure. The specimens are named as PCL-xBaTiO_3_, where x denotes the vol.% of BaTiO_3_ in the composites = 25, 45, and 65; we refer to PCL-xBaTiO_3_ as “PCL-xBT” for simplified nomenclature. The steps comprised one hour of stirring followed by another hour of ultrasonic dispersion and another two hours of stirring. Subsequently, the PCL-BT slurry was poured into glass Petri dishes and left in a laminar fume hood overnight for the dichloromethane to evaporate. The resultant was PCL-BT composite films (approximately 120 mm in diameter and 1.5 mm thick). These films were sheared into roughly 10 × 10 mm squares and made ready for filament extrusion (melt-blending). The squares were thoroughly dried in a desiccator before they were used for extrusion.

### 2.3. 3D-Printable Filament Development

The uniform diameter composite filaments were developed by a customized device assembly comprising (1) a single-screw extruder, (2) a sensor controlling the speed of filament winding, and (3) a motorized filament winding setup, as shown in Figure 2a. All the parts for the extrusion setup were procured from Filastruder (Snellville, GA, USA). The single screw extruder (with a Ø 2.75 mm nozzle) was kept at an elevated position with respect to the sensor and the winder setup. The sheared squares of the composite films were fed into the single-screw extruder, and the extrusion temperature and speed were set to 120 °C and 10 rpm, respectively. We used a fan just next to the nozzle, as shown in Figure 2a, to cool down the filaments. Once a continuous extrusion of uniform diameter filament was achieved, it was guided through the sensor. The sensor detects the filament’s position and adjusts the speed of the motorized winder in the filament winding setup. Usually, directly after the sensor, the filament is guided through a polytetrafluoroethylene (PTFE) tube loop onto the filament winding spool. The PTFE tube makes sure that there are no tangles in the filament. The motorized winding spool rotates at a specific speed (depending on the filament’s position as detected by the sensor), provides tension to the continuously extruded filaments, and helps form the uniform diameter 3D-printable composite filaments.

### 2.4. 3D Printing of Constructs

The custom-made filaments were loaded onto an FDM 3D printer (Funmat HT Enhanced, Intamsys) for developing the macroporous piezoelectric scaffolds, as shown in Figure 2b. The pore sizes of these microporous scaffolds were controlled by changing the infill density from 50–100%. The Simplify 3D software was used for slicing. For all the prints, a 0.4 mm nozzle was used. The print and bed temperatures were set at 130 and 30 °C, respectively. The layer height was maintained at 0.2 mm, and the print speed was 30 mm/s.

### 2.5. Material Characterization

Phase compositions of the specimens were detected using X-ray diffraction (XRD, Ultima III, RI, USA) with monochromated Cu Kα radiation (44 kV, 40 mA), focused beam mode over a 2θ range of 10–60°. Step width and count time were fixed to be 0.05° and 8 s during the analysis. A whole pattern fitting (WPF) analysis and Rietveld refinement were conducted using the MDI Jade software 2010 (Materials Data Inc., Livermore, CA, USA) to determine the goodness-of-fit. The morphology of the 3D-printed scaffolds was analyzed by a secondary electron detector in a Field Emission Scanning Electron Microscope (FESEM, Inspect F50, FEI, OR, USA). The cross-sections of the various 3D-printable PCL-BT filaments were also analyzed by FESEM.

### 2.6. Corona Poling and Electroactive Properties

The 3D-printed specimens were poled by the Corona Poling process to align the BaTiO_3_ dipoles in the composites (Figure 3a). We developed a customized setup, as shown in Figure 3b, to perform the corona poling process. Briefly, the specimens were kept on an aluminum plate that was grounded. The specimen was contained in an enclosed glass chamber, and a corona needle connected to the positive terminal of a high-voltage DC source was placed 3 mm away from the specimen surface. The DC voltage (8–12 kV) and the time (5, 15, and 30 min) were varied to identify the most optimum conditions for corona poling. Right after the poling, specimens were tested on a wide-range d_33_ meter (APC International, Ltd, Mackeyville, PA, USA) to measure the d_33_ coefficient and analyze the effect of corona poling on the electroactive properties of the samples. Specimens were also immersed in Phosphate Buffered Saline (PBS) for 14 days and then analyzed in a d_33_ meter.

### 2.7. Mechanical Properties

The mechanical testing specimens were 3D printed according to the dimensions mentioned in ASTM D638 (Figure 4). The same 3D printing parameters were followed as mentioned in Section 2.4 to develop/print the dog-bone-shaped tensile specimens. Subsequently, the specimens were mounted on a Universal Testing Machine (UTM). An Instron 3369 UTM with a 50 kN load cell was used for the tensile tests. Then, 5 mm/min strain deformation was used to measure the specimens’ tensile strength, and 1 mm/min strain deformation was used to measure the modulus, as per ASTM standards.

### 2.8. In Vitro Properties

Various PCL-BT were 3D printed into disc-shaped specimens (8 mm Ø × 2 mm height) for all the in vitro studies. Specimens were disinfected by immersing them in 90% ethyl alcohol for 30 min and then applying UV for another 10 min. Preosteoblast MC3T3 mouse cells (CRL-2593™, ATCC, Manassas, VA, USA) were used for the cell studies. The cells were cultured in complete minimum essential medium alpha (α-MEM, Thermo Scientific), supplemented with 10% fetal bovine serum (FBS, HyClone) and 1% penicillin/streptomycin (0.2 g/mL) at 5% CO_2_ and 37 °C.

#### 2.8.1. Routine Ultrasonic Stimulation (US) Application

For this assay, only PCL-65BT was chosen, as it was hypothesized that the latter would exhibit the highest electroactive properties. This assay was specifically designed to validate the effect of US stimulation on the piezoelectric scaffolds. The disc-shaped specimens were placed in 24-well plates and soaked with culture media. Then, ~2.3 × 10^4^ cells were directly seeded onto the specimens and allowed to attach to the scaffolds for 4 h. Subsequently, US was applied with the help of a clinical US setup (ComboCare Clinical Electrotherapy & Ultrasound Combo Unit). The probe of the US stimulator was attached to the bottom of the well plates, and the US was applied following a specific routine as shown in Figure 5. For validation purposes, US (1 or 3 MHz) was applied for 1 min continuously, every 3 h, two times/day for 72 h (3 days). At the end of 48 and 72 h, thiazolyl blue tetrazolium bromide (MTT) was used to determine the effect of US on cell proliferation if any. The cultured specimens were retrieved, rinsed three times in PBS, and prepared for the assay. The MTT solution was added to the specimens and control, followed by 4 h of incubation at 37 °C. Dimethyl sulfoxide was used to dissolve formazan, and finally, OD570 readings were recorded using a spectrophotometer.

#### 2.8.2. Cell Proliferation

The same US application procedure was followed for 3 or 7 days. At the end of 3 or 7 days, the specimens were retrieved, and the cell growth kinetics on those 3D printed piezoelectric specimens were determined using MTT assay as mentioned in Section 2.8.1.

#### 2.8.3. Cell Adhesion and Morphology

The 3D-printed specimens were soaked in α-MEM for 4 h and were seeded with ~6.5 × 10^4^ cells. The cell-seeded specimens were cultured for 24 h or 7 days, and at the end of the period, samples were retrieved, washed three times in PBS, and immersed in a 3% glutaraldehyde fixing solution for 1 h at room temperature. After fixation, they were dried sequentially in 30, 50, 70, 90, 95, and 100% ethyl alcohol, followed by a combined Hexamethyldisilazane (HMDS) and ethyl alcohol treatment. HMDS helps in chemically drying the specimens. Further, HMDS drying helps preserve the adhered cells’ structure on the specimens. Finally, they were dried overnight in a fume hood, plated with gold palladium for 60 secs, and imaged under FESEM.

#### 2.8.4. Cell Differentiation

In this assay, MC3T3 cells were seeded on multiple (more than the number needed for statistical analysis) specimens of the same composition. Then, 3 Mhz US was applied to the cells cultured on the scaffolds as shown in Figure 5. This is to maximize the amount of RNA collection. The cell-seeded specimens were cultured for 3 or 7 days, and at the end of specific times, RNA from these cells was extracted by TRIzol reagent (Invitrogen, Carlsbad, CA, USA). Complementary DNA was produced by reverse transcription using M-MLV reverse transcriptase (Promega, Madison, WI, USA), and quantitative Real-Time (qRT-PCR) was performed using SsoFast EvaGreen Supermix (Bio-Rad, Hercules, CA, USA) by a 2-step amplification program (30 s at 95 °C and 60 s at 62 °C) on a thermal cycler (Eppendorf, Hamburg, Germany). The relative quantification of mRNA of the gene of interest was determined by the 2^−ΔΔCT^ method and presented as fold change compared to the control sample (PCL). The forward and reverse primers for targeted genes are listed as follows:

Osteocalcin (OCN; forward 5′-GCAATAAGGTAGTGAACAGACTCC-3′ and reverse 5′-CTTTGTAGGCGGTCTTCAAGC-3′),

Osteopontin (OPN; forward 5′-CTTTCACTCCAATCGTCCCTAC-3′ and reverse 5′-GCTCTCTTTGGAATGCTCAAGT-3′),

Alkaline Phosphatase (ALP; forward 5′-ATCTTTGGTCTGGCTCCCATG-3′ and reverse 5′-TTTCCCGTTCACCGTCCAC-3′)

Type 1 Collagen (Col-1; forward 5′-GAGCGGAGTACTGGATCG-3′ and reverse 5′-GCTTCTTTTCCTTGGGGTT-3′), and

Runt-related transcription factor 2 (Runx2, forward 5′- CCAGATGGGACTGTGGTTACC3′ and reverse 5′-ACTTGGTGCAGAGTTCAGGG3′)

### 2.9. Statistical Analysis

All test results represented means ± standard deviation in triplicates. One-way and two-way analysis of variance (ANOVA) with the Tukey test was conducted to determine the statistical difference between groups, and *p* < 0.05 was considered significant.

## 3. Results and Discussion

### 3.1. Physical Characterization Results

Figure 6 shows the XRD plots of the various PCL-BT composite specimens. The unmodified PCL specimen exhibits sharp and well-defined peaks corresponding to the polymer. Specifically, two major peaks at the 2θ angles of 21.4° and 23.8° can be observed. These two peaks correspond to the (110) and (200) crystallographic planes of the semi-crystalline nature of PCL biopolymer, respectively [34,35]. However, incorporating BaTiO_3_ in the PCL matrix markedly reduced the intensity of the PCL diffraction peaks, and instead, new high-intensity and well-defined diffraction peaks were spotted. These peaks correspond to BaTiO_3_ as per JCPDS: #34-0129 [36,37]. The highest diffraction peak corresponding to BaTiO_3_ can be spotted at the 2θ angle of 30°, which corresponds to the (101) lattice plane. Furthermore, the other low-intensity yet sharp and well-defined peaks of BaTiO_3_ correspond to (111), (002), (112), (202), and (103) lattice planes [38]. Furthermore, the well-defined nature of the BaTiO_3_ diffraction peaks indicates the high crystallinity of the BaTiO_3_ particles. These findings are in accordance with previous studies that dealt with PCL- BaTiO_3_ composites, as mentioned in Ref [39].

The SEM micrographs in Figure 7a indicate the presence of uniform-sized and interconnected pores in the PCL-BT composite scaffolds. The average pore sizes of the scaffolds are mentioned in the image headers. Similar to our previous efforts in developing scaffolds with highly consistent geometry and uniform-sized pores, FFF stands out to be a highly efficient yet sustainable manufacturing technique for developing scaffolds with controlled pore sizes and geometry [19,40]. Previous studies focused on developing porous BaTiO_3_-based piezoelectric scaffolds using conventional methods such as freeze casting [36] or particulate (salt)-leaching [28]. However, even though such techniques can form porous scaffolds, there is little to no control over the pore size and geometry. Moreover, the probability of creating interconnected pores critical for cell nutrient flow might be less when using such conventional methods. As opposed, FFF is a powerful yet simple technique to develop scaffolds with highly consistent interconnected pores that should be leveraged to develop porous piezoelectric scaffolds [41].

The cross-sections of the various composite filaments are shown in Figure 7b. For composite materials, obtaining a homogenous dispersion of the secondary phase in the polymer matrix is critical to decreasing the chances of mechanical property degradation. The white dots in Figure 7b represent the BaTiO_3_ particles in the polymer matrix, which is seen as dark background, similar to previous reports [39]. The micrographs further indicate the homogenous distribution of the BaTiO_3_ microparticles in the PCL matrix, and no evident agglomerations were observed. As expected, the concentration of the white particles corresponding to the BaTiO_3_ particles increased with the higher incorporation percentage of the BaTiO_3_ in PCL. Importantly, even at 65 vol.% of BaTiO_3_ incorporation, there were no signs of BaTiO_3_ particle agglomeration, confirming that the combined steps of solvent casting and the two-step extrusion process are suitable for homogenous mixing of the BaTiO_3_ ceramic particles in the PCL polymer matrix. However, some BaTiO_3_ particle clusters were visible in the PCL-45BT and PCL-65BT composites, as identified by the red dotted circles in Figure 7b. The BaTiO_3_ cluster sizes were less than the agglomeration, similar to previous studies [42].

### 3.2. Piezoelectrical Properties

We poled the specimens to align the dipoles and to permanently achieve a prominent piezoelectric response. We specifically chose corona poling, as we wanted the specimens to be poled without immersing them in an oil bath, which might lead to eventual cytotoxicity in cells due to the remanent oil. We checked both the samples’ dry and wet piezoelectric responses, as the specimens are expected to be implanted in vivo, exposing them to bodily fluids. Hence, we immersed the specimens in PBS for prolonged periods for the wet piezoelectric response. We chose the PCL-25BT specimen to optimize our poling regime, hypothesizing that PCL-25BT would exhibit the lowest piezoelectric response among all the specimens and would be indicative of the piezoelectric effect. Figure 8a shows the dry piezoelectric response of the specimens. Poling at 8 and 10 kV for 5 and 15 min did not result in any piezoelectric response from the PCL-25BT, indicating that those voltages and times were insufficient to pole the specimens. However, the specimen yielded a piezoelectric response of 1.2 pC/N after poling at 10 kV for 30 min. We wanted to avoid prolonged poling times as we noticed the PCL-BT samples partially melted and deformed at the end of 30 min in cases of 12 and 15 kV due to the heat generated from the high voltages. Instead, the higher voltages of 12 and 15 kV for 5 and 10 min were sufficient to pole the samples with no deformation. The increase in poling voltage and time resulted in a gradual increase in the piezoelectric response of the specimens, but there were no significant differences between the specimen groups poled at voltages 12 and 15 kV. Furthermore, the specimens’ dry and wet piezoelectric responses are similar (Figure 8a,b). Overall, the specimens exhibited piezoelectric responses in the range of 1.2–2.25 pC/N with respect to different poling voltages and time. We chose our poling regime to be 15 kV for 15 min for further studies, as the PCL-25BT specimen exhibited a piezoelectric coefficient of 1.25 pC/N, similar to previous studies [42,43].

Figure 8c shows the piezoelectric responses of various PCL-BT specimens. The d_33_ piezoelectric coefficient of the PCL-BT composites increased with the increase in BaTiO_3_ content. Notably, the BaTiO_3_ inclusion up to 25 vol.% enhanced the piezoelectric response gradually to 1.2 pC/N compared to the unmodified PCL specimen; however, the piezoelectric response increased significantly when the BaTiO_3_ inclusion was above 25 wt.%. Specifically, the piezoelectric response increased to 2.4 and 2.6 pC/N for the PCL-45BT and PCL-65BT specimens. Several previous reports also observed similar results that dealt with BaTiO_3_ [36,42,44]. For instance, Liu et al. [42] observed a drastic rise in the d_33_ values (up to 3.9 pC/N) when the BaTiO_3_ incorporation extent increased above 35 vol.% in the PCL matrix. The drastic rise in the d_33_ coefficient is due to the dense distribution of the BaTiO_3_ particles in the PCL-45BT and PCL-65BT specimens as shown in Figure 7b. The electric field developed by the closely packed BaTiO_3_ particles forms a network interacting with each other, resulting in a more robust electroactive response. On the contrary, the BaTiO_3_ particles are sparsely distributed in the PCL-25BT specimens (Figure 7b), resulting in reduced electroactive responses.

The d_33_ piezoelectric coefficient of the PCL-BT specimens in the present study is similar to the results of the other BaTiO_3_-based composites previously reported in the literature. For instance, porous HA-BaTiO_3_ composites explored by Zhang et al. [36] exhibited d_33_ values in the range of 0.3–2.8 pC/N. Tang et al. [44] recorded d_33_ values of polarized HA-BaTiO_3_ piezoelectric ceramics in the range of 1.3–6.8 pC/N with BaTiO_3_ content ranging from 80% to 100%. Some of the HA-BaTiO_3_ composites with less than 80 vol.% BaTiO_3_ did not exhibit any piezoelectric effect, highlighting the need for high-volume BaTiO_3_ contents in the composites [45,46]. On the other hand, some of the bulk-sintered HA-BaTiO_3_ composites exhibited outstandingly high d_33_ values (>50 pC/N) because of the close packing density of the piezoelectric BaTiO_3_ particles in a sintered sample [47]. Polymer–ceramic composites such as the PCL-BT can never exhibit such a high piezoelectric response because the BaTiO_3_ particles are not densely packed in a polymer matrix. Moreover, such a high scaffold-mediated piezoelectric response is not needed for bone regeneration, as the bone itself exhibits a piezoelectric response in the range of 0.7–2.3 pC/N [48]. The 3D-printed PCL-BT composites in the present study exhibited d_33_ values that are close to the piezoelectric response of bone and were also on par with HA-BaTiO_3_ composites developed by various conventional techniques [36,43,44]. Hence, the observed piezoelectric response of the PCL- BaTiO_3_ composites in the present study highlights their great potential in the bone remodeling process and osteogenesis and expediting treatment of orthopedic defects.

### 3.3. Mechanical Properties

Figure 9a,b shows the mechanical properties of the piezoelectric PCL-BT specimens. It is evident that the inclusion of 25 vol.% BaTiO_3_ particles increased the modulus by 35% (Figure 9a) and the tensile strength by 14% (Figure 9b). Similar results were observed by Bagchi et al. [39] when the authors incorporated various kinds of perovskite ceramic nanoparticles, including BaTiO_3_, in the PCL matrix. However, in the latter study, the authors observed an increase in the yield strength after adding 20 wt% BaTiO_3_ in PCL, even though the increase was the lowest (about 10%) when compared to the other ceramic additives explored in that study. As opposed, in the present study, we observed a decrease in the modulus and the tensile strength when the BaTiO_3_ inclusion extent was increased above 25 wt.%. For instance, increasing the BaTiO_3_ incorporation percentage to 45 and 65 vol.% reduced the modulus by 9% and 15%, respectively, with respect to unmodified PCL. Furthermore, the tensile strength decreased from 14 MPa (for unmodified PCL) to almost 12 MPa for PCL-45BT and 11 MPa for PCL-65BT specimens (Figure 9b) However, the tensile strain % of the PCL-BT composites was not influenced by the content of the BT particles. All the composites exhibited a high tensile strain % with an average value of 550%, similar to PCL specimens, which exhibited a strain of 590%.

It is well known that the incorporation of excess secondary phases in polymer composites can hamper the polymer’s inherent mechanical properties. Moreover, including ceramic particles in polymers can increase the brittleness of the composites [23,24]. The excess secondary materials can form agglomerations in the matrix, disrupting the efficient load/stress transfer between the ceramic particles and the matrix. However, we did not notice a massive decrease in the mechanical properties with higher BaTiO_3_ contents (>25 vol.%), indicating that the mixing and distribution of the BaTiO_3_ particles were favorable. However, the decrease in the mechanical properties of the PCL-45BT and PCL-65BT specimens was due to the clustering of the BaTiO_3_ particles, as shown in Figure 7b.

All the PCL-BT specimens in the present study were developed by FFF-based 3D printing. In FFF, the printed layers are glued to each other layer by layer to develop the final material; this process of product manufacturing can significantly influence the mechanical properties of the manufactured parts, especially when the print parameters and orientations are incorrect. Even though optimizing the 3D-printing parameters to achieve a robust 3D-printed part was not the scope of this study, our extensive experience in 3D printing of various polymers and composites [18,19,20,23,24,40] helped us achieve mechanically robust 3D-printed parts. To highlight, the 3D-printed PCL-25BT specimen’s mechanical properties were similar to the PCL-BT specimen (with 20 wt.% BaTiO_3_) that was developed by conventional compression molding [39]. Overall, the mechanical properties of the PCL-BT scaffolds qualify them as mechanically robust scaffolds for orthopedic applications.

### 3.4. Piezoelectric Effect on the in Vitro Properties of Pre-Osteoblasts

While performing a literature review for this study, we observed that most of the studies did not apply any stimuli to the piezoelectric scaffolds during the in vitro cell studies, such that the scaffolds would exhibit the piezoelectric response during the cell studies. In most cases, the studies used poled samples for cell studies, which raises the question, “*Whether it is the piezoelectric effect from the scaffolds or is it the surface charge on the scaffolds that is helping in cellular activities?*”. We were specifically interested in analyzing the scaffold-mediated piezoelectric effect on the cells; hence, we chose ultrasonic stimulation (US) as the stimuli to create the deformation on the scaffolds and exhibit the piezoelectric effect. Several studies in the literature used US to stimulate BATiO_3_ 3=-based piezoelectric scaffolds and confirmed that US is a highly effective stimulus to enhance the piezoelectric effect in electroactive scaffolds [17,49]. Moreover, we chose US, as it does not physically come in touch with the cell-laden scaffolds avoiding contamination issues.

#### 3.4.1. Pre-Osteoblast Proliferation

##### Effect of US and Piezoelectricity on Pre-Osteoblast Proliferation

Figure 10a,b shows the in vitro cell proliferation properties of the MC3T3 cells cultured on the PCL-65BT piezoelectric scaffolds. We chose PCL-65BT as our test specimen, as the latter specimen exhibited the highest piezoelectric effect, and we hypothesized it would have a prominent effect on the pre-osteoblast cells. First, we wanted to analyze the short-term impact of the US and scaffolds’ piezoelectric stimulation on the pre-osteoblasts proliferation behavior. As shown in Figure 10a, we observed that US alone had no positive influence on cell proliferation. However, 1 MHz US combined with the piezoelectric scaffolds significantly increased cell proliferation over 48 and 72 h with respect to the only US treatment. The 3 MHz US treatment, combined with the piezoelectric scaffolds, was also beneficial, but the treatment did not result in a significant increase in cell proliferation. The results were similar when the US regime continued for up to 7 days, as shown in Figure 10b. However, after the 7-day treatment and culture, the 3 MHz US treatment combined with the piezoelectric scaffolds yielded nearly similar results to the 1 MHz US, indicating that over long treatment times, the 1 and 3 MHz US application regimens can deliver similar outcomes in terms of pre-osteoblast proliferation. Notably, the cells proliferated significantly over 7 days, irrespective of the specimen and the treatment. Nevertheless, we determined that the 1 MHz US treatment and the piezoelectric stimulation from the PCL-65BT scaffolds are optimum for pre-osteoblast proliferation. Once we determined the optimum US regimen for cell proliferation, we next analyzed the effect of BaTiO_3_ on cell proliferation.

##### Effect of Piezoelectric BaTiO_3_ on Pre-Osteoblast Proliferation

Figure 11a confirms that the BaTiO_3_ presence in all the composite scaffolds significantly increased the cell growth kinetics over time as compared to the non-piezoelectric PCL scaffolds. This is due to the increased piezoelectric response resulting from the BaTiO_3_ content in the PCL-BT composites. However, even though there was a significant difference in the extent of cell proliferation as compared to the PCL scaffolds, there was no significant difference in the extent of cell proliferation among the various PCL-BT specimens. The SEM images (Figure 11a,b) of the adhered pre-osteoblast cells on various 3D-printed scaffolds corroborate the proliferation results. Very few cells adhered to the scaffolds after day 1 of the culture (Figure 11a). In addition, many circular and spindled cells were seen, indicating that these cells did not have time to spread over the surface with their filopodia. However, after day 7 of the culture, it is evident that the cells proliferated notably and spread over the scaffolds with extending filopodia, as shown in Figure 11b. Furthermore, it is evident that the increasing BaTiO_3_ concentration in the composites results in enhanced cell adhesion and spreading. In all the cases, the cells approvingly adhere and spread on the surface with flattened morphology, signifying that the PCL-BT scaffolds exhibit a bioactive and biocompatible surface. The increasing cell adherence and spread over the composites with increasing BaTiO_3_ concentration demonstrate that the scaffold-generated piezoelectric effect played a major role in cell spreading [50]. The increasing BaTiO_3_ particles enhance the piezoelectric effect, which helps the cells to proliferate and spread more rapidly on the scaffold surfaces. However, even though more cells proliferated and were seen to adhere on the PCL-45BT and PCL-65BT specimens as compared to PCL-25BT specimens, there was not a significant difference in the amount of adhered cells on all the composites. Similar results were obtained by Tang et al. [44], where the authors observed no significant difference in the osteoblast proliferation on the composite scaffolds containing different contents of BaTiO_3_ (80 and 90 wt% BaTiO_3_). Similarly, Tavangar et al. [46] reported no significant difference in cell adhesion and proliferation between the composite scaffolds with varying BaTiO_3_ (40, 50, and 60% wt BaTiO_3_). This could be possible because the BaTiO_3_ particles are embedded in the polymer or ceramic matrix, and in most cases, they are not directly interacting with the cells; hence, the variation in BaTiO_3_ presence does not play a significant role in cell proliferation [51,52]. Instead, we believe that the extent of the piezoelectric effect generated from the BaTiO_3_ particles influences cellular activities more prominently than the biocompatibility nature of the BaTiO_3_.

Clearly, US alone does not affect cellular activities in the present case. However, US helped in remarkable cell proliferation when the piezoelectric scaffolds were present, indicating that the US invigorated the scaffolds and generated the piezoelectric effect, increasing the cell growth kinetics. Furthermore, the increase in BaTiO_3_ content resulting in the enhancement of pre-osteoblast proliferation is a direct indication that US can interact with the BaTiO_3_ particles and generate the piezoelectric effect. The effect increases when US interacts with more BaTiO_3_ particles in the composites, notably surging the cell growth kinetics for the PCL-65BT specimens. Shuai et al. [53] developed PVDF-BaTiO_3_ scaffolds and demonstrated that the electric cues generated by the piezoelectric scaffolds under US efficiently promoted MG-63 cell proliferation. Yang et al. [17] observed similar results when they used US on PLLA- BaTiO_3_ and PLLA-BaTiO_3_-graphene scaffolds and deduced that the US was responsible for yielding effective piezoelectric response from the scaffolds. Even BaTiO_3_ coatings on metallic scaffolds and US have remarkably increased bone cell activity. For instance, Fan et al. [49] coated BaTiO_3_ onto the porous titanium (Ti) scaffold and applied low-intensity pulsed US (LIPUS), similar to the clinical US used in the present study, and observed that LIPUS was highly influential in activating the piezoelectric effect of BaTiO_3_. The latter effect significantly enhanced cell viability in vitro, promoted osteogenesis and osseointegration in vivo, and effectively treated large bone defects in a rabbit model. The same research group performed another study in a large animal model (sheep) and confirmed that LIPUS induces a piezoelectric response from BaTiO_3_ particles, leading to an increased cellular response in vitro and treating sizeable segmental bone defects in vivo [54]. Recently, Chen et al. [50] also used LIPUS on the BaTiO_3_-coated Ti scaffold and coined the new term “piezodynamic therapy” for such kinds of treatment. The authors observed that the piezodynamic effect of US and piezoelectric BaTiO_3_ coating activated more mitochondria at the initial stages of cell culture that intervened in the cell culture cycle by promoting cell proliferation and weakening the apoptotic damage. These studies reinforce that US combined with piezoelectric PCL-BT scaffolds has a strong potential to enhance bone cell activities and help in new bone formation and defect regeneration in vivo.

##### Effect of Pore Size on Pre-Osteoblast Proliferation

We also observed that the pore sizes significantly impacted the cell proliferation kinetics. For instance, as shown in Figure 12, 3D-printed PCL-BT scaffolds with 70% infill (avg pore size: 320 µm) resulted in the highest cell growth compared to the other macro-porous scaffolds and the non-porous scaffolds. This is because the 320 µm pore size provides an optimum distance for cell-cell communication as opposed to other pore sizes. Similarly, Lee et al. [55] reported 350 µm pore sizes to be the optimum for cell proliferation. However, porous scaffolds can compromise the mechanical properties of load-bearing orthopedic scaffolds. In one of our recent studies, we observed that 3D-printed scaffolds with 300 µm pore size exhibit the highest yield compressive strength, and increasing the pore size beyond that would decrease the specimen’s yield strength [19]. The 320 µm pore-sized scaffolds in the present study resulted in the highest cell proliferation, indicating that such PCL-BT scaffolds would be beneficial for increasing cell growth kinetics and exhibit favorable mechanical properties.

Bone tissue engineering has extensively utilized porous scaffolds to enhance implant–tissue interaction. The osteogenic capability of the orthopedic scaffold can be enhanced significantly by interconnected pores, as they facilitate cell distribution, nutrients, and blood flow. Furthermore, porous scaffolds help in robust anchorage, scaffold-tissue osseointegration, and vascularization [56,57]. However, conventional manufacturing techniques such as salt leaching, gas foaming, phase separation, and freeze-drying do not control pore sizes and numbers. In addition, the formed pores lack interconnectivity, which is critical for nutrient flow, cell migration, vascularization, and tissue ingrowth. Furthermore, scaffolds with random pores and wide variations in pore sizes weaken the scaffold’s mechanical strength or load-bearing capacity, thus leading to implant failure. In contrast, 3D printing is a powerful manufacturing technique that can form uniform-sized interconnected pores in scaffolds, facilitating osteogenesis and osseointegration. Thus, the 3D-printed PCL-BT scaffolds developed in the present study will not only help bone cell proliferation via piezoelectricity but will also help expedite new bone formation with the help of the highly uniform pores.

#### 3.4.2. Osteoblast Differentiation

Figure 13a–e show the differentiation behavior of the MC3T3 pre-osteoblasts over time when exposed to the US treatment and the piezoelectric scaffolds. As per previous reports, we selected the 3 MHz US for the differentiation studies and observed that the 3 MHz treatment effectively differentiates pre-osteoblast cells into osteoblasts. All the piezoelectric PCL-BT scaffolds upon US application increased the mRNA expressions notably of all the osteogenic gene markers not after 3 days, but after 7 days of culture. This is usual, as the gene markers explored in this present case are observed in stages of osteoblastic differentiation, and it usually takes longer than 3 days for the MC3T3 pore-osteoblasts to differentiate [58,59,60,61]. Notably, the PCL-45BT and PCL-65BT specimens markedly increased the ALP, Col-1, OPN, OCN, and Runx-2 expressions, which are critical in osteogenesis.

The results in the present study are very similar to previous studies. For instance, Shuai et al. [53] observed no effect of only US on the ALP expressions and osteogenic differentiation of the MG-63 cells. However, ALP activities of the cells significantly increased when US was applied to the piezoelectric PVDF- BaTiO_3_ scaffolds. Similarly, Fan et al. [49] observed significantly higher mRNA gene expression of Col-1, ALP, OPN, and Runx2 when BaTiO_3_ coatings and LIPUS were used complementarily after 7 and 14 days. In the present study, we observed upregulation of all the gene markers only after 7 days. As mentioned before, the gene expression of ALP, Col-1, and Runx2 usually appears in the early stages of osteogenic differentiation of bone cells, while OPN and OCN expression occurs late in the process of osteogenic differentiation. Moreover, the US enhanced the mechanical–electrical conversion capability (piezoelectric response) of the scaffolds and yielded more electrical charges. It is proven that electrical charges activate the electrically sensitive calcium ions (Ca^2+^) signal transduction [10], and Ca^2+^ plays a critical role in regulating osteoblast differentiation via either activation of the calcium-sensing receptors or increasing the Ca^2+^ influx into the osteoblast cells [62]. Thus, the electrical stimulation (due to the piezoelectric effect) from the PCL-BT scaffolds activated the electrically sensitive Ca^2+^ signal transduction, activating the calcium-sensing receptors and increasing the Ca^2+^ influx into MC3T3-E1 cells. The latter series of events resulted in a notable increase in osteogenic gene expression, specifically ALP [63,64]. The remarkable rise in ALP level in MC3T3 cells will result in enhanced bone matrix and mature extracellular matrix synthesis, as observed by Tian et al. [63]. Moreover, in an important recent study, Cai et al. [65] observed a remarkable increase in ALP, OCN, Col-1, and Runx-2 expression when they applied US to BaTiO_3_-coated Ti scaffolds. The authors proved that the piezoelectric effect from the scaffolds specifically upregulated the mRNA and protein expressions of the calcium channel protein Ca_V_1.2 in MC3T3 cells, which was responsible for the notable osteogenic differentiation.

Furthermore, Zhou et al. [66] indicated that the charged substrate surface helps in adsorbing proteins and various biological constituents from the culture medium through electrostatic interaction, which might also be another reason for promoting cell attachment, proliferation, and differentiation. Thus, the US successfully triggered an adequate piezoelectric response from the PCL-BT scaffolds, resulting in the upregulation of prominent osteogenic-related gene expressions.

On a separate note, electrical fields are instrumental in cell migration during wound healing [3,67]. Even though analyzing cell migration under the piezoelectric effect was not the scope of this study, we believe that the adhered cells on the scaffolds might have migrated toward the surface regions, which were devoid of cells (similar to a wound), eventually covering the entire scaffold as confirmed by the SEM images in Figure 12b,c. We draw a resemblance that the blank regions on the scaffold at the beginning of the culture (i.e., after day 1) are similar to a wound that is usually devoid of cells. Eventually, by the end of 7 days, the cells approvingly proliferate and spread over the blank surfaces on the scaffolds with the generation of the piezoelectric effect from the scaffolds. Morimoto et al. [68] observed that the migration distance was significantly longer, and the area of the transplanted bone cells was markedly wider in an ischemic stroke model of rats when they applied electrical stimulation. Such studies confirm the various effects of scaffold-mediated electrical fields/electrical charges/electrical stimulation on bone cell activities.

Finally, the combination of the US and PCL-BT scaffolds enhanced the bone cell activities, highlighting that the US could interact with the embedded BaTiO_3_ particles in the 3D-printed composite scaffolds. Our results strongly confirm that US can be an effective stimulus to enhance the piezoelectric effect in the PCL-BT scaffolds, indicating that the US-activated piezoelectric response from electroactive PCL-BT scaffolds can be a powerful therapeutic tool for enhanced osteogenesis and expedited bone defect or fracture treatment.

## 4. Conclusions

Overall, this study has three significant findings to highlight. First, this study calls attention to the strong utilization of polymer–ceramic multi-functional 3D-printable filaments. We successfully formed PCL-BT 3D-printable filaments that can be utilized in an FFF setup to seamlessly develop design-specific scaffolds that can lead to the clinical translation of various customized, multi-functional implants for reconstructive applications in the field of orthopedics and beyond. Even though it is beneficial to develop multi-functional implants, few efforts have been made to form polymer–ceramic composite filaments for FFF. This study provides evidence that composite filaments can be leveraged to create multi-functional medical implants and devices. Second, this study highlights the utilization of FFF-based 3D printing to develop design-specific piezoelectric and regenerative implants. Few efforts have been made to 3D print piezoelectric implants focused on regenerative medicine; most efforts are concentrated on piezoelectric sensors. Instead, this study provides evidence that 3D printing and innovative biomaterial compositions can be leveraged to develop regenerative piezoelectric scaffolds not only for orthopedic applications but also for maxillofacial, cranial, and dental applications. Such scaffolds can also be considered in a digital workflow for implantology [69]. Finally, this is one of the few studies to prove that clinical US treatment combined with piezoelectric scaffolds can be instrumental in expediting bone regeneration and cure and help minimize the treatment time in patients with severe orthopedic injuries. However, follow-up in vivo studies is mandatory to further validate the efficacy of the piezoelectric implants in terms of EF generation at the implantation site and the extent of bone regeneration.

## Figures and Tables

**Figure 1 bioengineering-09-00679-f001:**
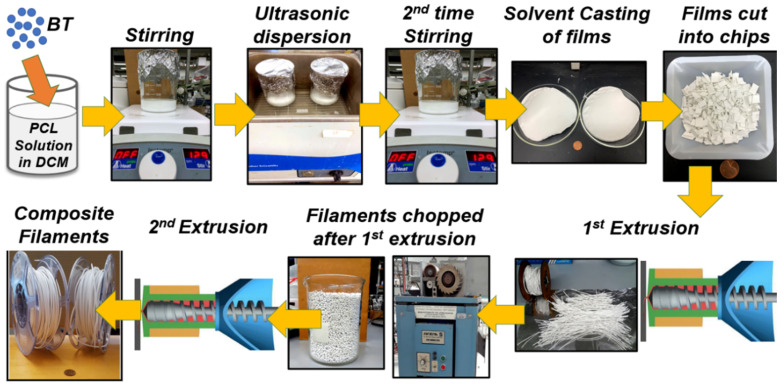
Composite filament development. Schematics show the entire workflow of developing the polycaprolactone–barium titanate (PCL-BT) composites and the 3D-printable PCL-BT filaments.

**Figure 2 bioengineering-09-00679-f002:**
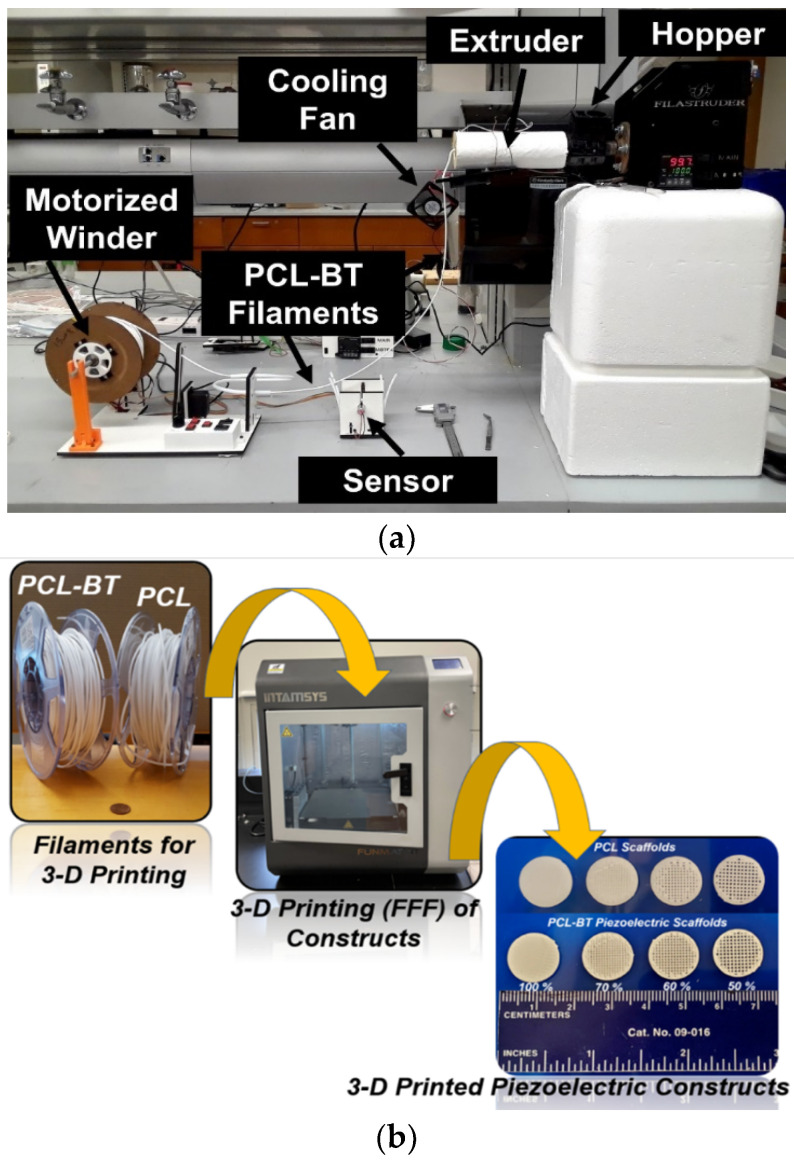
Setup for filament extrusion and 3D printing. (**a**) The assembled Filastruder extruder and filament winder setup that was used to extrude the 3D-printable filaments. (**b**) The workflow shows that the filaments were used in a 3D printer setup to develop the piezoelectric PCL- BT composite scaffolds.

**Figure 3 bioengineering-09-00679-f003:**
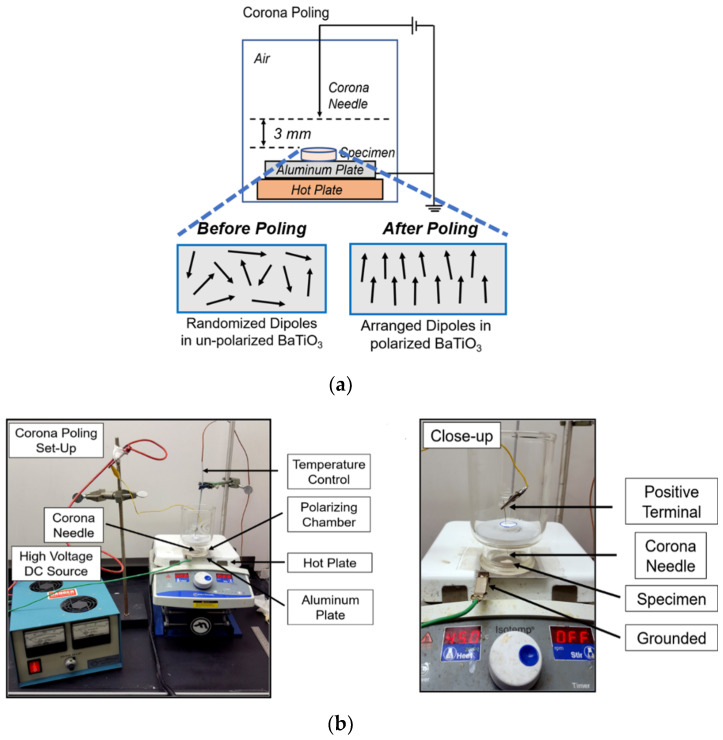
Polarization of the PCL-BT scaffolds. (**a**) Schematics of the Corona Poling setup. (**b**) Digital image of the coronal poling setup.

**Figure 4 bioengineering-09-00679-f004:**
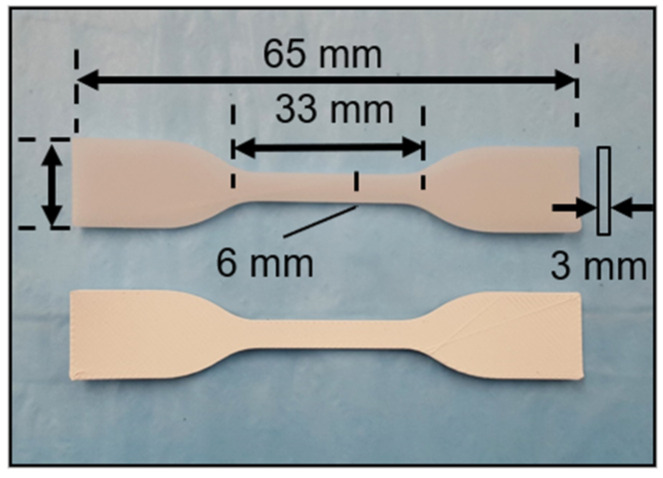
Digital image of the 3D-printed PCL and PCL-45BT scaffolds.

**Figure 5 bioengineering-09-00679-f005:**
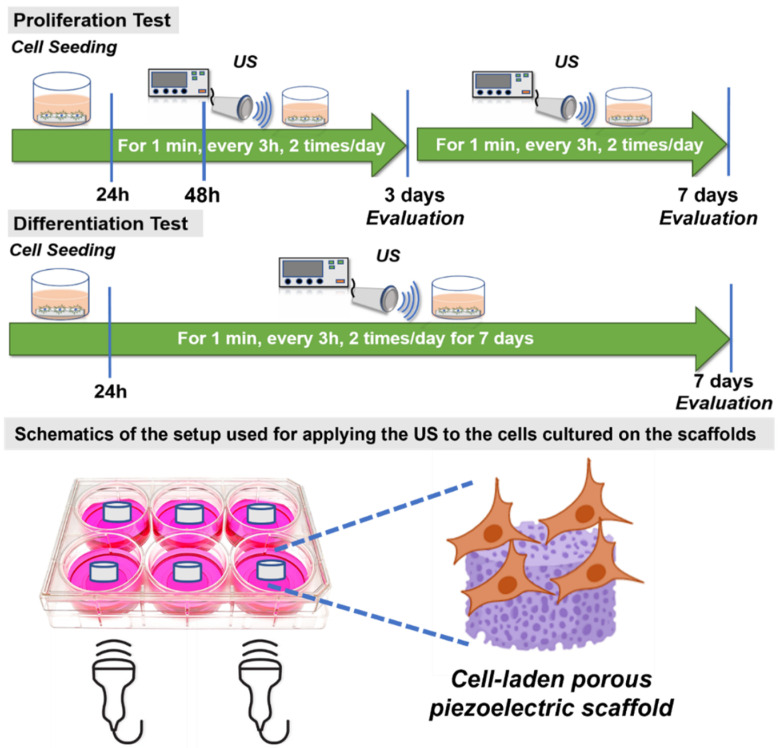
Schematics showing the routine for applying the US on the PCL-BT piezoelectric scaffolds.

**Figure 6 bioengineering-09-00679-f006:**
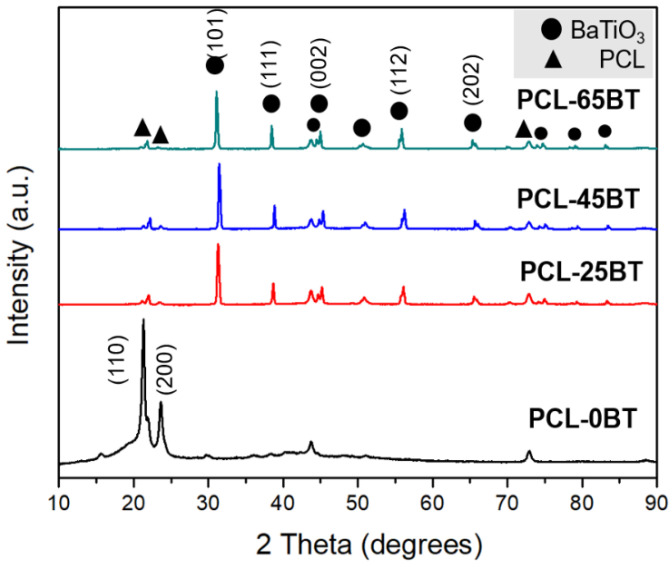
X-ray diffraction (XRD) of the PCL-BT composite scaffolds with various concentrations of BaTiO_3_.

**Figure 7 bioengineering-09-00679-f007:**
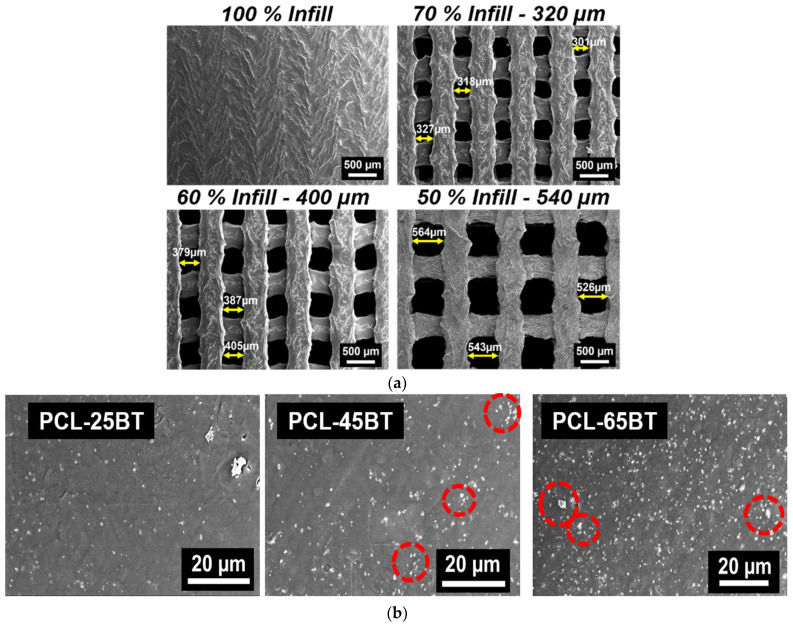
SEM analysis of the scaffolds using secondary electron detector. (**a**) SEM of the non-porous and porous 3D-printed PCL-45BT scaffolds. The pore sizes of the scaffolds were varied by changing the infill percentage in the slicing software. (**b**) SEM of various PCL-BT filament cross-sections showing the uniform dispersion of the white BaTiO_3_ particles in the PCL matrix.

**Figure 8 bioengineering-09-00679-f008:**
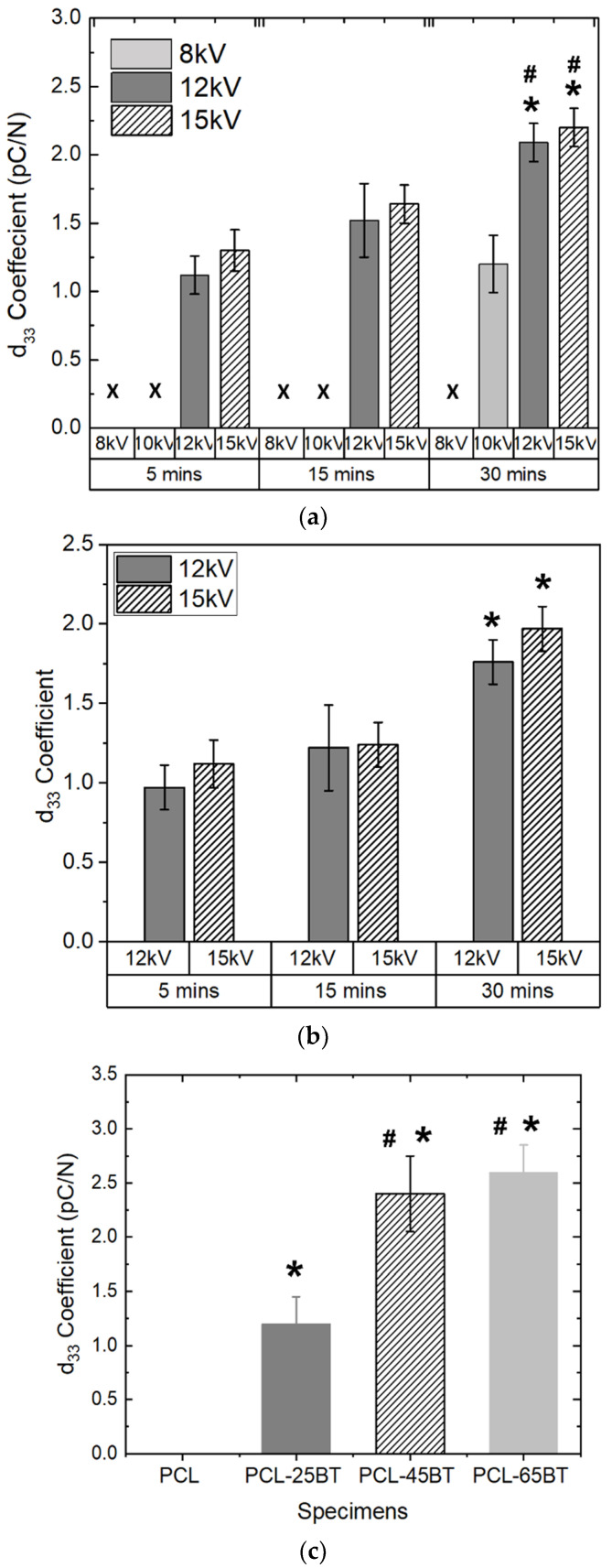
d_33_ coefficient (picoColoumbs/Newton, pC/N) of (**a**) dry and (**b**) wet PCL-BT scaffolds with respect to the poling voltage and time. # means statistically significant (*p* < 0.05) with respect to the specimen poled at 10 kV in the same group (30 min). * means statistically significant (*p* < 0.05) with respect to the specimens poled for 5 and 15 min. (**c**) d_33_ coefficient of various wet PCL-BT scaffolds. * means statistically significant (*p* < 0.05) with respect to PCL. # means statistically significant (*p* < 0.05) with respect to PCL-25BT.

**Figure 9 bioengineering-09-00679-f009:**
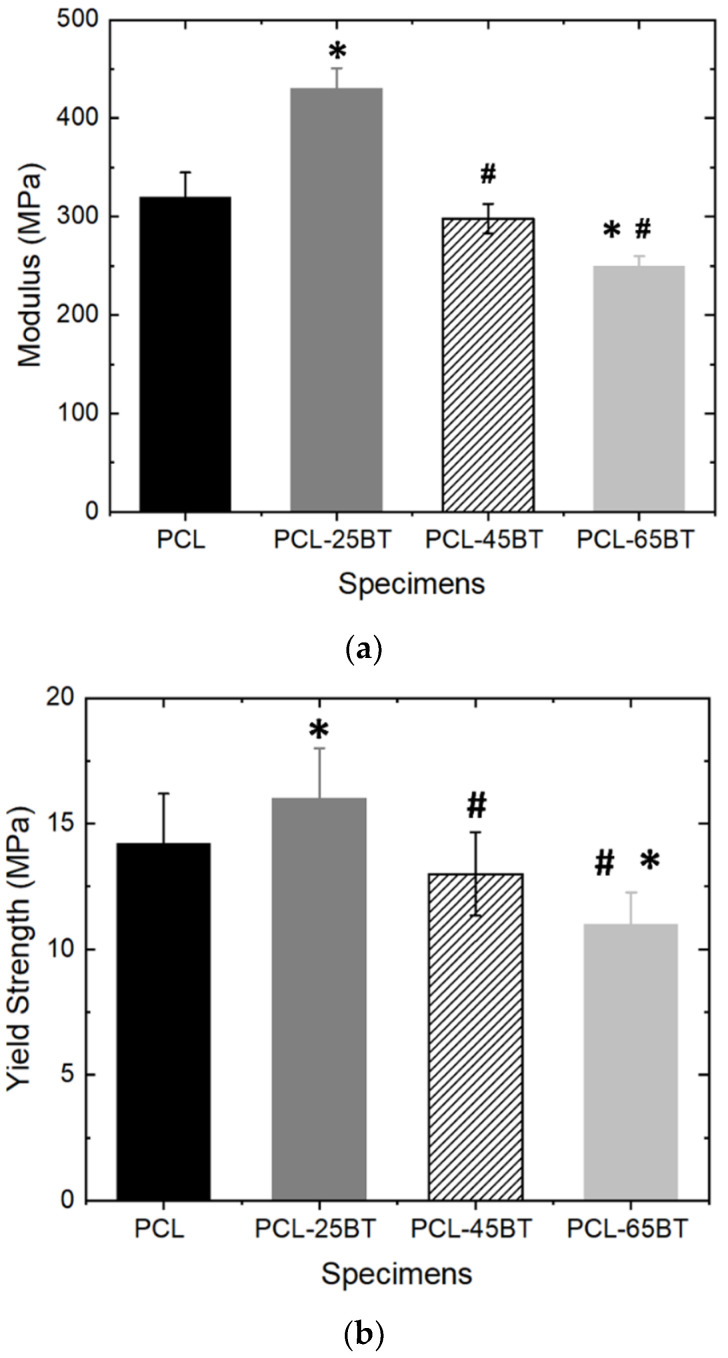
Mechanical testing of the composite specimens. (**a**) Young’s modulus and (**b**) yield tensile strength of the 3D-printed PCL and PCL-BT scaffolds. * means statistically significant (*p* < 0.05) with respect to PCL. # means statistically significant (*p* < 0.05) with respect to PCL-25BT.

**Figure 10 bioengineering-09-00679-f010:**
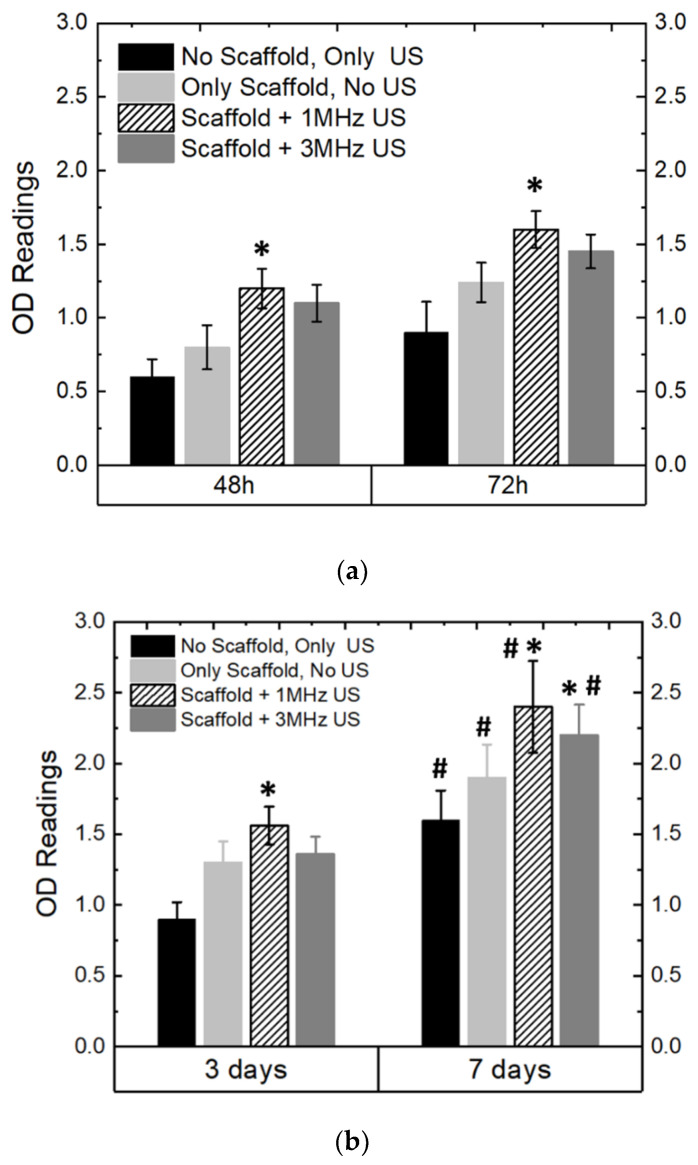
Ultrasonic stimulation (US) and in vitro properties of the scaffolds. OD readings of MC3T3 cells as cultured on the piezoelectric scaffolds (PCL-65BT) with respect to different intensities of US over (**a**) 72 h and (**b**) 7 days. * means statistically significant (*p* < 0.05) with respect to “no scaffold, only US”. **#** means statistically significant (*p* < 0.05) with respect to the specimens in the “3 days” group.

**Figure 11 bioengineering-09-00679-f011:**
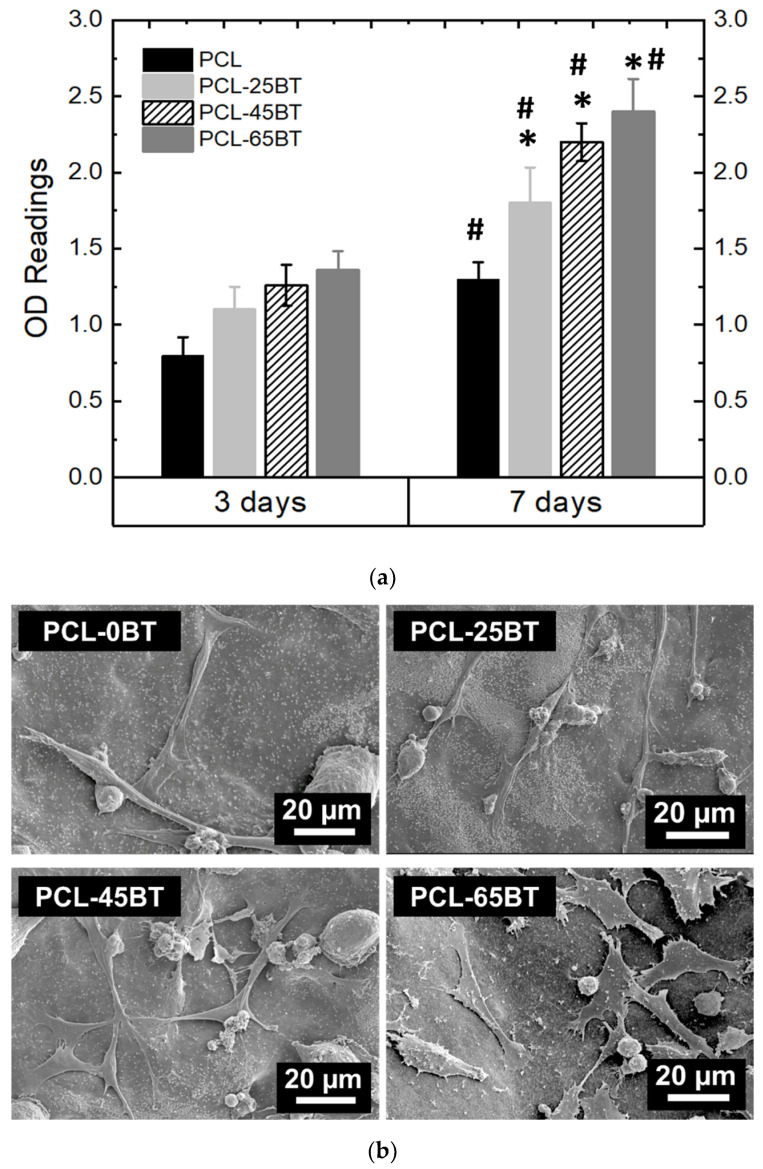
Effect of BaTiO_3_ content on the in vitro properties of the piezoelectric scaffolds. (**a**) OD readings of the MC3T3 cells cultured on the PCL and various piezoelectric PCL-BT scaffolds; 1 MHz US were used per the routine for all the proliferation test. * means statistically significant (*p* < 0.05) with respect to PCL specimens. **#** means statistically significant (*p* < 0.05) with respect to the specimens in the “3 days” group. SEM images of the adhered MC3T3 cells on various scaffolds (**b**) after day 1 and (**c**) after day 7 of culture.

**Figure 12 bioengineering-09-00679-f012:**
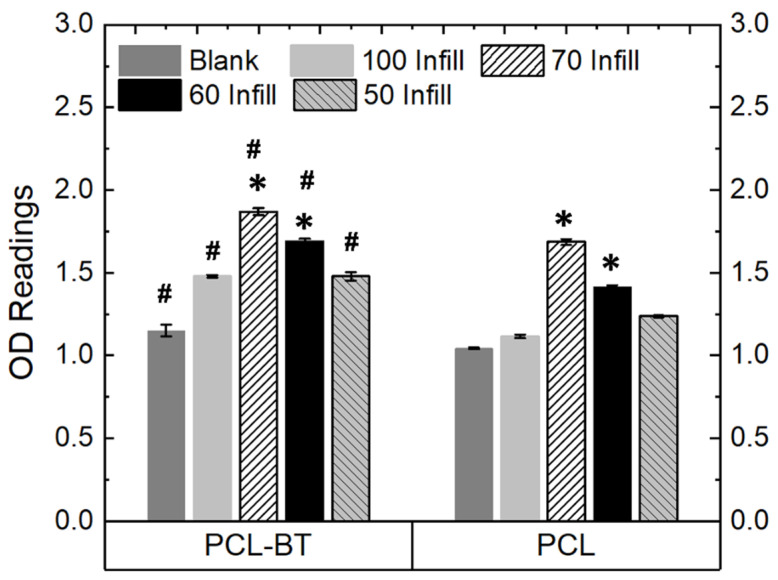
OD readings of MC3T3 cells cultured on the piezoelectric scaffolds (PCL-65BT) with different infill (pore sizes); 1 MHz US was used per the routine for all the proliferation tests. * means statistically significant (*p* < 0.05) with respect to the controls (Blank and 100 Infill specimens) in the same group. # means statistically significant (*p* < 0.05) with respect to the PCL specimens with different infills.

**Figure 13 bioengineering-09-00679-f013:**
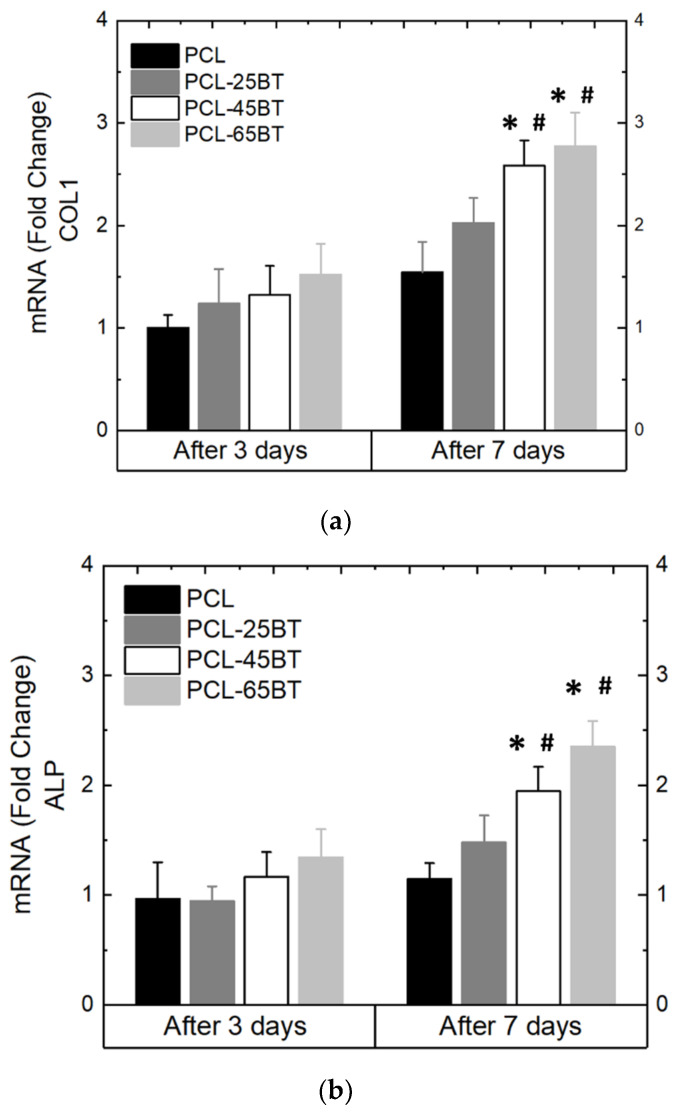
Differentiation properties of MC3T3 cells as cultured on the piezoelectric scaffolds. The following osteogenic gene markers were quantified: (**a**) Col 1 (**b**) ALP (**c**) OCN, (**d**) OPN, and (**e**) Runx 2 and expressed as fold change with respect to the control; 3 MHz US was used per the routine for all the differentiation assays. * means statistically significant (*p* < 0.05) with respect to the control (PCL) in the same group. **#** means statistically significant (*p* < 0.05) with respect to the fold changes of the same specimen in the “after 3 days” group.

## Data Availability

Data will be made available on request.

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
