# Peer review of "3D-Printed Piezoelectric Porous Bioactive Scaffolds and Clinical Ultrasonic Stimulation Can Help in Enhanced Bone Regeneration"

_bioengineering, 2022, doi:10.3390/bioengineering9110679_

Round 1

Reviewer 1 Report

The subject is somewhat clear, and it has been explored much more than the current introduction gives credit. The article presents a good idea.  Although the initial question is interesting, I have a few issues with the study.

This article fits into the framework of translational orthopaedics: how to fill the gap between basic sciences and clinical sciences. How to understand  bone regeneration Stenosis using Bioactive Scaffolds and Clinical Ultrasonic Stimulation. The author can add a paragraph in the introduction to explain the idea in the context of translational medicine. Example of an article should be cited:

doi: 10.1080/03007995.2017.1385450

doi: 10.1097/BCO.0000000000000846

1. Title: good
2. Abstract: it captures the appropriate essence of the manuscript. Excellent.

3. Introduction: The introduction identifies the problem that is being addressed in the manuscript and doesn't state the purpose of the manuscript. Add at the end: In this article, we will explain..., second, ....
4. Tables and figures: Quality of figures is so important too. Thanks for providing this good resolution.

5. References: I have verified all references and all key references are correct. Please update the reference according to the article proposed above.
6. Methods: the authors validate the idea with experimental study. I appreciate it .

7. Discussion:

* The authors don't discuss the  limitations of the study. Please add it.
8. Conclusion: The conclusion is justified by the methods and results.
9. There are still some mistakes in grammar and misprints, the authors should carefully check this manuscript.

* I have enjoyed reading, and I am in favor of publication after suitable.

Author Response

Dear Reviewer,

We would like to express our thankfulness and gratitude to you for putting your valuable time into intricately reviewing this manuscript. We are happy to receive your expert comments that enhance the manuscript's quality. We have considered all the comments, and the answers to those are mentioned in the following.

In the revised manuscript, the changes are highlighted in yellow.

We really appreciate all the positive comments from you. Thank you for all the positive and inspiring comments. In the following, we just address the comments where you have suggested making modifications.

Comment 1: The author can add a paragraph in the introduction to explain the idea in the context of translational medicine. Example of an article should be cited:

doi: 10.1080/03007995.2017.1385450

doi: 10.1097/BCO.0000000000000846

Response: We agree with the reviewer. In the revised manuscript, we have included a paragraph about translational medicine. We have cited the papers mentioned by the reviewer. The paragraph reads as –

We believe that the result of this proof-of-concept study is the first step towards developing an effective orthopedic-based regenerative medicine for the society. EFs have been used in clinics for wound healing for several years. Furthermore, the components/materials of the orthopedic scaffold used in this study are FDA-approved, making the translation process relatively easy. Indeed more effective collaboration between clinician scientists, engineers, orthopedic surgeons, and federal funding agencies is required to conduct clinical trials and translate innovative medical devices such as this piezoelectric polymer-ceramic scaffold for expedited bone regeneration.

Comment 2: The article only indicates the possibility of printing and crosslinking a material that has not been tested for biocompatibility (even essential) since it has been designed for use in tissue engineering as indicated by the authors. Therefore the work needs to be completed with some biocompatibility tests, also bearing in mind that gelatin and alginate are not the most suitable materials for future use in clinical applications.

Response: Towards the end of the Introduction, we have added the purpose of the manuscript. It reads –

… in this study, we developed a unique electroactive and bioactive polymer-ceramic composition in the form of 3D printable filaments such that the filaments could be utilized in a FFF 3D printing setup to manufacture design-specific piezoelectric orthopedic scaffolds. To achieve that, first, we introduced micron-sized piezoelectric BT particles in the polycaprolactone (PCL) polymer matrix by solvent-casting and subsequently used a customized extrusion process to form uniform diameter 3D printable PCL-BT composite filaments. Then, the filaments served as feedstock material for a FFF set up to develop the design-specific, porous scaffolds. We chose PCL because it is an FDA-approved, biodegradable polymer that has been extensively used in implantable biomaterials, and BT because of its well-known piezoelectric and biocompatible properties. After the scaffold development, we performed thorough material, mechanical, and in vitro properties analyses of the 3D printed structures. The phase composition and morphology of the implants were determined by X-ray diffraction and Scanning electron microscopy (SEM). Mechanical properties of the composites were determined using ASTM standards, and the proliferative and differentiative capacity of the scaffolds was assessed in vitro using MC3T3 pre-osteoblast cells.

Comment 3: The authors don't discuss the limitations of the study. Please add it.

Response: Thank you for the suggestion. We added the following part at the end of the manuscript in the conclusion to highlight the missing experiments of this study–

Finally, this is one of the few studies to prove that clinical US treatment combined with piezoelectric scaffolds can be instrumental in expediting bone regeneration and cure and help minimize the treatment time in patients with severe orthopedic injuries. However, follow-up in vivo studies is mandatory to further validate the efficacy of the piezoelectric implants in terms of EF generation at the implantation site and the extent of bone regeneration. 

Comment 4: There are still some mistakes in grammar and misprints, the authors should carefully check this manuscript.

Response: We have thoroughly checked the manuscript for mistakes in grammar and typos. We have also professional software to check the grammar of the entire manuscript.

Reviewer 2 Report

Dear authors,

please consider the following comments for imrpoving your manuscript:

-always a dash between 3D and printing, pe 3D-printed. Correct throughout the text where needed.

-the chemical compounds don't need capital letters, pe. polycaprolactone, barium titanate, fused filament fabrication . Correct throughout the text, where needed. BT is not proper, choose the chemical formula for short writing (likewise l. 70-72)

-consider removing the abbreviations that are not often used pe EF, US

-", sensors, and actuators": never use a comma before an "and" and "or". Check and correct throughout the text

-"neo-bone formation": consider an more formal expression or replace with " "neo-bone" formation"

-" β-poly": greek letters in italics font,  β-poly, 2θ 

-l. 64-63: correct font colour

-replace: "polyamide-11 (PA-11)"

-replace: "poly(lactic acid-co-glycolicacid)"

-remove DCM for dichloromethane. The chemical compounds have formulae, not abbreviations! And no capital letters are needed for their names, except when they induce a new sentence.

-replace: 15 %w/v

-replace: 10 mm, 120 oC, always a space between the numbers and the units. Check and correct throughout the text

-squares: are they vessels or items with the shape of the polymer extruded?

-rewrite the captions of Fig. 1, 2 (format issue, capital letters)

-what is the use of the abbreviations cited in Introiduction, if the full names are used in Experimental and Results too? 

-check and correct subtitle 2.3

-Fig. 5c,d,e: reshape the size of the charts, too big. Apart from the experimental apparatus set, the charts that refer to results should be placed later in text, not in part 2.

-"min" instead of minutes, "mL" instead of ml: check and correct throughout the text

-font colour l. 653-659

-check reference style if follows the journal's instructions

Author Response

Dear Reviewer,

We would like to express our thankfulness and gratitude to you for putting your valuable time into intricately reviewing this manuscript. We are happy to receive your expert comments that enhance the manuscript's quality. We have considered all the comments, and the answers to those are mentioned in the following.

Comment 1: Always a dash between 3D and printing, pe 3D-printed. Correct throughout the text where needed..

Response: Thank you for the suggestion. We have replaced the word as ‘3D-printed’.

Comment 2: The chemical compounds don't need capital letters, pe. polycaprolactone, barium titanate, fused filament fabrication. Correct throughout the text, where needed. BT is not proper, choose the chemical formula for short writing (likewise l. 70-72)

Response: Thank you for the suggestion. We corrected it throughout the test. Specifically, we have used the formula of Barium Titanate throughout the manuscript. But in many places, we prefer to use PCL-BT as the nomenclature just for simplicity in mentioning/labeling them in the figures.

Comment 3: consider removing the abbreviations that are not often used pe EF, US

Response: Thank you for the suggestion. We are not removing the abbreviations of EF and US, because they have been used in the manuscript several times in the manuscript.

Comment 4: ", sensors, and actuators": never use a comma before an "and" and "or". Check and correct throughout the text

Response: Thank you for the suggestion. The grammatical checking professional software suggests otherwise. However, we have tried to remove the comma before “and” and “or”.  

Comment 5: "neo-bone formation": consider an more formal expression or replace with " "neo-bone"

Response: Thank you for the suggestion. We have rectified the text as per the suggestion.

Comment 6: -" β-poly": greek letters in italics font,  β-poly, 2θ 

Response: Thank you for the suggestion. We have rectified the text as per the suggestion.

Comment 7: -l. 64-63: correct font colour

Response: Thank you for the suggestion. We have rectified the text as per the suggestion.

Comment 8: replace: "polyamide-11 (PA-11)"

Response: Thank you for the suggestion. We have rectified the text as per the suggestion.

Comment 9:  replace: "poly(lactic acid-co-glycolicacid)"

Response: Thank you for the suggestion. We have rectified the text as per the suggestion.

Comment 10 : remove DCM for dichloromethane. The chemical compounds have formulae, not abbreviations! And no capital letters are needed for their names, except when they induce a new sentence.

Response: Thank you for the suggestion. We have rectified the text as per the suggestion.

Comment 11: replace: 15 %w/v

Response: Thank you for the suggestion. We have rectified the text as per the suggestion.

Comment 12: replace: 10 mm, 120 oC, always a space between the numbers and the units. Check and correct throughout the text

Response: Thank you for the suggestion. We have rectified the text as per the suggestion.

Comment 13: squares: are they vessels or items with the shape of the polymer extruded?

Response: They are scaffolds that have been printed.

-rewrite the captions of Fig. 1, 2 (format issue, capital letters)

Response: Thank you for the suggestion. We have rectified the text as per the suggestion.

-what is the use of the abbreviations cited in Introduction, if the full names are used in Experimental and Results too? 

Response: Thank you for the suggestion. We have now used the abbreviations once defined in the Introduction in the rest of the manuscript.

-check and correct subtitle 2.3

Response: Thank you for the suggestion. We have rectified the subtitle as per the suggestion.

-Fig. 5c,d,e: reshape the size of the charts, too big. Apart from the experimental apparatus set, the charts that refer to results should be placed later in text, not in part 2.

Response: Thank you for the suggestion. We will upload all the source files to the journal.

-"min" instead of minutes, "mL" instead of ml: check and correct throughout the text

Response: Thank you for the suggestion. We have rectified the text as per the suggestion.

-font colour l. 653-659

Response: Thank you for the suggestion. We have rectified the text as per the suggestion.

-check reference style if follows the journal's instructions

Response: Thank you for the suggestion. Yes, we have checked the reference style.

Reviewer 3 Report

The paper is very interesting

However, before acceptance, some topics are needed to be addressed:

1) Such scaffold can be useful in the case of full-arch implant rehabilitation? Can it represent an aid in treating diabetic patients? Please cite PubMed ID36142007

2) Such scaffold can be useful to get a better outcome of implant prosthetic rehabilitations, both soft and hard tissues? Please cite DOI10.23805/JO.2018.10.04.043) Such scaffold can be considered in a digital workflow for implantology? please cite PubMed ID34425664

Author Response

Dear Reviewer,

We would like to express our thankfulness and gratitude to you for putting your valuable time into intricately reviewing this manuscript. We are happy to receive your expert comments that enhance the manuscript's quality. We have considered all the comments, and the answers to those are mentioned in the following.

In the revised manuscript, the changes are highlighted in yellow.

Comment 1: Such scaffold can be useful in the case of full-arch implant rehabilitation? Can it represent an aid in treating diabetic patients? Please cite PubMed ID36142007

Response: Yes, such scaffolds can be used in full-arch implant rehabilitation and also help in treating diabetic patients. We have cited the mentioned reference. The included text reads as follows –

Notably, scaffold-mediated expedited bone regeneration can be beneficial for patients with diabetes who have a compromised or delayed wound healing and tissue regeneration rate

Comment 2: Such scaffold can be useful to get a better outcome of implant prosthetic rehabilitations, both soft and hard tissues? Please cite DOI10.23805/JO.2018.10.04.043) Such scaffold can be considered in a digital workflow for implantology? please cite PubMed ID34425664

Response: Yes, the scaffold can be useful to get a better outcome of implant prosthetic rehabilitations, both soft and hard tissues, and also can be considered in a digital workflow for implantology We have cited the article PubMed ID34425664 requested. But the DOI10.23805/JO.2018.10.04.043 cannot be found.  

Round 2

Reviewer 2 Report

Corrections have been overseen:

-Always a dash between 3D and printing, pe. 3D-printed, 3D-printing, 3D-printable etc. Check and correct throughout the text where needed

-Minimize a bit the size of some figures so no great gaps are shown in pages. Be careful regarding the figures' captions or the numeration (Fig. 5a, b,c etc) Follow the instructions of the template

-at some points the font colour is grey instead of black, correct

-check the reference format at the end, if follows the instructions

-The chemical compounds don't need capital letters, pe. polycaprolactone, barium titanate, fused filament fabrication. Correct throughout the text, where needed. BT is not proper, choose the chemical formula for short writing (likewise l. 70-72, boron nitride etc)

Be attentive of your manuscript so it results in a good paper.
